# Antimicrobial and Mechanical Properties of Ag@Ti_3_C_2_T_x_-Modified PVA Composite Hydrogels Enhanced with Quaternary Ammonium Chitosan

**DOI:** 10.3390/polym15102352

**Published:** 2023-05-18

**Authors:** Linxinzheng Guo, Kun Hu, Haibo Wang

**Affiliations:** 1Beijing Engineering Research Center of Printed Electronics, Institute of Printing and Packaging Engineering, Beijing Institute of Graphic Communication, Beijing 102600, China; guodazheng045@163.com (L.G.); Wanghaibobbo@163.com (H.W.); 2Collage of Biological Science and Engineering, Fuzhou University, Fuzhou 350108, China

**Keywords:** MXene, 2-hydroxypropyltrimethylammonium chloride chitosan, mechanical properties, antibacterial activity, strain

## Abstract

Polyvinyl alcohol (PVA) is a polymeric material with good biocompatibility, excellent hydrophilicity, and a large number of hydroxyl groups. However, due to its insufficient mechanical properties and poor inhibition of bacteria, it has a lack of applications in wound dressings, stent materials, and other fields. In this study, a simple method was used to prepare composite gel materials: Ag@MXene-HACC-PVA hydrogels with a double-network structure were prepared using an acetal reaction. Due to the double cross-linked interaction, the hydrogel has good mechanical properties and is resistant to swelling. The adhesion and bacterial inhibition were enhanced due to the addition of HACC. In addition, the strain sensing properties of this conductive hydrogel were stable, and the GF (specification factor) was 1.7617 at 40–90% strain. Therefore, the dual-network hydrogel with excellent sensing properties, adhesion properties, antibacterial properties, and cytocompatibility has potential applications in biomedical materials, especially as a tissue engineering repair material.

## 1. Introduction

Every year, countless people have their organs damaged due to trauma, war, accidents, etc. Tissue engineering technology provides new methods for organ trauma repair and functional reconstruction [1,2,3]. The preparation of an ideal material is also one of the focuses of researchers [4,5].

PVA is one of the synthetic polymers commonly used in biomaterials in recent years. Due to its inherent good biocompatibility, it has been widely used in the field of tissue engineering and biosensing [6]. However, PVA hydrogel as a biomaterial also has some disadvantages, such as poor mechanical properties, poor bacterial inhibition, and poor adhesion, which lead to the easy growth of bacteria on the hydrogel surface. Hydrogel as a bionic skin, while promoting the proliferation of fibroblasts, accelerates the growth of bacteria, causing bacterial infection during tissue engineering repair, and the consequences are even life-threatening. Various methods have been used to improve the mechanical and biological properties of hydrogels (antibacterial properties, biocompatibility, etc.). For example, tough hydrogels with multisized layered structures were prepared by changing multiple promoted hydrogen bonds using freeze casting [7], and the bacterial inhibition rate was improved by the incorporation of antibacterial molecules [8,9,10,11]. However, a hydrogel with a simple preparation, strong mechanical properties, and good adhesion and antibacterial properties is still sought by researchers.

Over the years, due to the emergence of graphene materials [12], there has been an increasing interest in two-dimensional nanomaterials, especially MXene [13]. Due to its large specific surface area and good mechanical, electronic, and physicochemical properties, as well as its excellent biocompatibility, MXene is widely used in optoelectronics, environmental science, sensors, and biomedicine [14,15,16,17,18,19]. All MXene nanosheets contained in MXene-based gels were obtained by using different top-down fluorine-containing etching methods to selectively remove the “A” layer (i.e., mainly group 13 and 14 elements) from its parent 3D MAX phase [20]. For the wide application of hydrogels, MXene has provided new methods for the development of hydrogels by virtue of its abundant functional groups (-OH, -O, etc.) and conductive properties, giving them multifunctionality [21]. For example, Rasool et al. [22] introduced the antibacterial activity of colloidal Ti_3_C_2_T_x_-modified films, and the hydrophilic MXene coating showed good antibacterial activity against both Escherichia coli and Bacillus subtilis. Liu et al. [23] doubled MXene and PANI to enhance the electrical conductivity and tensile properties of PVA-based hydrogels and relied on MXene to provide the antibacterial ability of near-infrared light irradiation in their preparation. Li et al. [24] prepared anisotropic MXene/PVA hydrogels with good electrical conductivity and high tensile strength and bacterial inhibition by using a directional freeze-assisted salting method, which not only had excellent mechanical properties (stress up to 0.5 MPa, strain up to 800%) but could also be used for local thermal treatment of infected sites with an NIR laser (808 nm). In addition, the researchers carried out the self-reduction reaction of the antibacterial particles with MXene on MXene nanosheets, and the antibacterial effect was improved compared with pure MXene. Li et al. [25] physically cross-linked Ag/MXene with PVA via borax to prepare antibacterial and sensing hydrogels, and the reaction of MXene with boric acid ions (B(OH)_4−_) weakened the strength of the PVA–borax cross-linked network. Ag/MXene could be used as a conductive and antibacterial component, resulting in hydrogels with good electrical conductivity and antibacterial activity (93% inhibition of *S. aureus* and 95% inhibition of *E. coli*) and improved hydrogel elongation (580%). Wang et al. [26] assembled positively charged chitosan and negatively charged Ag@MXene on the PLLA surface by the layer-by-layer self-assembly (LBL) method, and the growth inhibition of both *E. coli* and *S. aureus* under 808 nm NIR laser radiation was higher than that of pure MXene-PLLA membrane. In addition, MXene has good electrical conductivity and has made great progress in sensing. Gou et al. [7] prepared an artificial ear membrane based on an MXene (Ti_3_C_2_T_x_) acoustic sensor with polydimethylsiloxane (PDMS) as the substrate, which could achieve speech detection and recognition. Yang and Qin [27,28] both used MXene composite polymer materials to prepare sensing hydrogels to achieve high sensitivity and a low response time, which have wide application prospects in wearable medical applications. However, there are few articles on the cross-linking of Ag@MXene with PVA by acetal reaction for hydrogels and improving mechanical and biological properties on this basis.

Considering the above-mentioned situation, this study aimed to achieve bacterial inhibition with the addition of Ag@MXene to PVA hydrogels and to improve their adhesion, mechanical properties, etc., as biomedical materials. Here, the use of 2-hydroxypropyltrimethylammonium chloride chitosan (HACC) was considered. Water-soluble quaternary ammonium salts produced by the reaction of chitosan with trimethylammonium chloride of glycidyl esters as chitosan derivatives have better water solubility and film-forming properties with enhanced antibacterial ability against Staphylococcus aureus. Quaternized chitosan has been touted as having a significant antimicrobial active agent [29]. As the most typical natural cationic modified chitosan, it has the advantages of easy availability, low cost, antibacterial activity, and biocompatibility [30,31]. 2-hydroxypropyltrimethylammonium chloride chitosan was added to the PVA-Ag@MXene ionic cross-linking system to prepare hydrogels with good mechanical strength, biocompatibility, strain sensing, adhesion, and antibacterial properties. In this paper, the surface morphology, swelling behavior, mechanical properties, adhesion, antibacterial properties, biocompatibility, and strain sensing properties of Ag@MXene-HACC-PVA hydrogels were systematically investigated. The results indicated that Ag@MXene-HACC-PVA hydrogels have good potential applications as tissue engineering materials.

## 2. Materials and Methods

### 2.1. Materials

Ti_3_AlC_2_ powder was purchased from Nanjing Pioneer Nano. Polyvinyl alcohol (PVA, MW = 3 × 10^4^ − 7 × 10^4^) and glutaraldehyde (GA) were purchased from Aladdin (Shanghai, China). Silver nitrate (AgNO_3_), lithium fluoride (LiF), and hydrochloric acid (HCL) were purchased from Sinopharm Chemical Reagent Co. 2-hydroxypropyl trihydroxyammonium chloride chitosan (HACC; degree of substitution ≥ 90%; prepared from CS with MW = 161.16) was purchased from Wuhan Lanabai Pharmaceutical Chemical Co., Ltd, Wuhan, China. All reagents were of analytical grade. The purchased chemical reagents could be used directly without further physicochemical treatment, and the experimental water was deionized water. Mouse fibroblasts (L929 cells) were obtained from the General Hospital of the Chinese People’s Liberation Army. CCK-8 experiments were performed with the Invigentech Cell Counting Kit, Shanghai, China. Live/dead cell staining kits were purchased from Shanghai Bestbo for fluorescent staining of L929 live/dead cells. *E. coli* and *S. aureus* strains were provided by State Key Laboratory of Pathogen and Biosecurity (Beijing Institute of Biotechnology).

### 2.2. Ag@MXene Colloidal Solution Preparation

Ag@MXene (Ag@Ti_3_C_2_T_x_ composite) was prepared as described in our previous study [26]. First, Ti_3_AlC_2_ powder was slowly added to the etching solution containing LiF and hydrochloric acid and stirred at 50 °C for 24 h. Then, it was washed with deionized water until pH = 6 and dried. MXene colloidal solution was prepared by sonication in ice water under argon atmosphere and centrifuged at 3500 rpm for 1.5 h. MXene colloidal solutions were prepared by sonication in ice water under argon atmosphere and centrifugation. Finally, AgNO_3_ solution was slowly added dropwise to the MXene colloidal solution and stirred for sonication to prepare Ag@MXene colloidal solution with 20% silver content.

### 2.3. Preparation of Ag@MXene-PVA-HACC Hydrogels

Ag@MXene hydrogels were prepared by acetal reaction according to the available reports. First, the colloidal solution was slowly dropped into 13.5 mL of deionized water and sonicated for 10 min to make a uniform mixture. Then, HACC powder was slowly added to the above solution and stirred at 50 °C for 2 h. Then, PVA powder was added to it and stirred at 90 °C for 6h to dissolve it completely. The above solution was cooled to room temperature, and HCl (1M) was slowly added dropwise until the solution’s pH was 3. Finally, an appropriate amount of glutaraldehyde was added and poured into the mold and fully cross-linked at 40 °C for 2 h to obtain Ag@MXene-PVA-HACC hydrogel with a mass ratio of PVA to GA of 1:1.

### 2.4. Material Characterization

The chemical structure of Ag@M-H-PVA hydrogels was analyzed in attenuated total reflection (ATR) mode using an FTIR spectrometer (FT-IR 100, WI, USA) at room temperature. Prior to the measurements, the hydrogels were purified by immersion in deionized water and freeze-dried under vacuum. SEM and EDS energy spectra were observed using a scanning electron microscope (ZEISS Gemini SEM 300, Oberkochen, Germany). The electrical conductivity of Ag@M-PVA/Ag@M-H-PVA hydrogels with different contents was measured using a four-probe square resistance tester (ROKO Riker Micro, FT-331, Shanghai, China). The tensile properties of PVA, Ag@M-PVA, and Ag@M-H-PVA hydrogels and the adhesion strength of Ag@M-H-PVA hydrogels were measured using a high- and low-temperature electronic tensile machine (5565A, INSTRON, Boston, MA, USA) at room temperature. For the measurement of tensile properties, the hydrogels were dumbbell-shaped with a thickness of 2 mm, a width of 5 mm, and a length of about 45 mm. The tensile speed was set to 10 mm/min. For the assessment of the adhesion strength of the hydrogels, the hydrogels were cut into 25 mm × 25 mm squares, and the connecting plates on the apparatus were pulled at a speed of 10 mm/min until they separated. The adhesion strength was calculated by dividing the measured maximum load value by the area of the adhered sample. The swelling behavior of the hydrogel was assessed by weighing the mass of the hydrogel before and after immersion using an electronic balance. Motion monitoring was performed by directly attaching the hydrogel at the finger, wrist, throat, and balloon; stretching using a modular flexible test system (FlexTes-s-P2, Hunan, Beijing, China); and recording resistance changes with an LCR meter (TH2830, Changzhou, China) at 1 V AC voltage and 1 kHz sweep, which was used to evaluate the hydrogel strain properties.

### 2.5. In Vitro Antibacterial Activity Assay

The inhibition ability of Ag@MXene-PVA and Ag@MXene-PVA-HACC hydrogels against *E. coli* and *S. aureus* was evaluated using the plate count method. First, the hydrogels were immersed in 4 mL of 10^8^ CFU mL^−1^ bacterial suspension (220 rpm, 37 °C, 4 h), and the control group was the sample-free group. The samples were incubated in a thermostatic shaker at 37 °C for 4 h. Subsequently, 100 μL of the dilution solution (tenfold dilution) was evenly spread on agar plates, and the incubated colonies were counted after incubation for 18 h in an incubator at 37 °C. Meanwhile, to compare the effect of photothermal synergy, each group was irradiated under an 808 nm NIR laser (1.5 W/cm^2^) for 15 min. Materials and bacteria were incubated and counted as above. The inhibition ability of *E. coli* and *S. aureus* was assessed in the same way. The inhibition rate was calculated as follows in Equation (1):(1)$$X=\frac{A_{0}-A_{1}}{A_{0}}\times 100\%$$
where X is the inhibition rate, A_0_ is the average number of colonies of the control sample, and A_1_ is the average number of colonies of the sample under test.

### 2.6. In Vitro Biocompatibility Assay

CCK-8 was used to detect the change in cell viability of PVA, Ag@M-PVA, and Ag@M-H-PVA hydrogels when incubated with L929 for 24 h and 72 h, and cell survival was observed by live/dead staining. An appropriate amount of a sample was weighed, sterilized by UV irradiation for 30 min, and added with serum medium (100 μg/mL). A total of 5% CO_2_ was extracted in a constant-temperature incubator at 37 °C for 3 days. A 0.22 μm filter membrane was filtered to remove bacteria, and the extract solution (100 μg/mL) was obtained. Cells were cultured in medium containing 10% fetal bovine serum, penicillin (100 μg/mL), streptomycin (100 μg/mL), and amphotericin (0.25 μg/mL). L929 cells in logarithmic growth phase were taken, the cells counted, and the cell concentration adjusted, and the cells were inoculated into 96-well plates with 6 × 10^3^/well and 5% CO_2_ and incubated overnight in a constant-temperature incubator at 37 °C to adhere the cells to the wall. The cells were incubated for 1 and 3 days using the above extracts. The absorbance values at 450 nm were detected using an enzyme marker. The calculation of relative cell viability using measured OD values was as follows (2):(2)$$\mathrm{Cell}\;\mathrm{viability}\%=\frac{{\mathrm{OD}}_{\exp}-{\mathrm{OD}}_{\mathrm{bla}}}{{\mathrm{OD}}_{\mathrm{con}}-{\mathrm{OD}}_{\mathrm{bla}}}\times 100\%$$
where OD_bla_ is the OD value of the blank group, OD_con_ is the OD value of the control group, and OD_exp_ is the OD value of experimental groups.

For live/dead staining experiments, similar to the CCK-8 method described above, cells were inoculated in 24-well culture plates and PBS-washed once in the 24-well plates to remove excess serum. Cells were stained at 1 d and 3 d. The staining was terminated by washing three times with PBS, and the proliferation of cells was observed using a laser confocal microscope and photographed.

### 2.7. Statistical Analysis

All experimental groups were estimated as at least 3 times the mean standard deviation (x + SD, n = 3,4,5...). Statistical analysis was performed using Student’s *t*-test or one-way analysis of variance. * *p* < 0.05, ** *p* < 0.01, *** *p* < 0.001 were considered statistically significant.

## 3. Results and Discussion

### 3.1. Formation Mechanism

Multifunctional Ag@M-H-PVA hydrogels were prepared in one step using the acetal reaction. The preparation process of the hydrogels and the schematic diagram of the photothermally enhanced antibacterial properties are shown in Figure 1a. To demonstrate the formation of Ag@M-H-PVA hydrogels, FTIR spectra of the hydrogels (HACC3wt%, Ag@MXene2wt%) are shown in Figure 1b. This shows the stretching vibration peak of the N-H bond at 3310.29 cm^−1^, the stretching vibration of the -OH absorption peak at 3300~2500 cm^−1^, the stretching vibration of CH_2_ at 2917.13 cm^−1^, and the stretching vibration of C=O at 1700 cm^−1^. The C-N stretching vibration in HACC is 1184 cm^−1^; C-O stretching vibration is 1300–1100 cm^−1^; and C-H bending vibration absorption is 1000–600 cm^−1^. The -CHO vibration in glutaraldehyde generated by 2820–2720 cm^−1^ disappears. Comparing the state before and after the reaction, the successful polymerization of the hydrogel was also confirmed by curing into a smooth-surfaced hydrogel after heating. The surface uniformity of the hydrogel was observed using EDS, as shown in Figure 1d. Carbon (C) mainly follows the surface morphology of the hydrogel, which is consistent with the stoichiometric morphology presented by the SEM images. Silver (Ag) and titanium (Ti) presence confirmed that the Ag@MXene nanosheets were abundantly distributed in the hydrogel, but titanium was clearly visible as aggregates in some places, presumably due to the inhomogeneous sonication of the Ag@MXene colloidal solution during the addition of deionized water, which allowed Ag@MXene to aggregate on the hydrogel surface. Oxygen (O), as a common element present in both components, showed a uniform distribution.

The hydrogel system contains PVA molecular chains with hydroxyl groups, MXene containing hydroxyl groups, HACC molecular chains with positive charges, AgNPs with bacteriostatic effects, and GA molecular chains containing aldehyde groups after free-radical-initiated polymerization. In polymers, there are three types of interactions, which include interactions by hydroxyl groups in PVA and MXene chains, PVA and PVA chains, HACC and PVA chains, HACC and MXene chains, HACC and HACC chains, and MXene and MXene chains, forming weak hydrogen bonding interactions; hydroxyl groups in any molecular chain that interact with aldehyde groups in GA molecular chains for acetal reactions, forming interactions with strong covalent bonds; and electrostatic interactions generated by MXene and HACC. Based on these structural features of the hydrogel network, the strong covalent interactions help to enhance their elasticity and mechanical strength (toughness and fracture strength), and the weak hydrogen interactions help to dissipate mechanical energy and obtain good tensile properties [32]. Moreover, the introduction of silver nanoparticles piggybacking MXene confers high inhibition, electrical conductivity, and high strain sensitivity to the hydrogels.

### 3.2. Mechanical Properties

Firstly, the effects of different contents of Ag@MXene on the mechanical properties of the Ag@M-PVA hydrogels were investigated. Tensile tests were conducted on the PVA hydrogels and Ag@M-PVA hydrogels. In Figure 2a, the stress–strain curves of the PVA hydrogels and Ag@M-PVA hydrogels with different contents are shown. The results show that the maximum tensile stress of the pure PVA hydrogel was 8.0825 KPa and the breaking strain was 16.976%, while the tensile strength of the hydrogel after the addition of Ag@MXene decreased in a cliff-like manner, and the maximum tensile stress in the hydrogel containing 1 wt% Ag@MXene was 0.35595 KPa with a strain at break of 2.353%. This is because the addition of Ag@MXene breaks the dynamic equilibrium within the original hydrogel, resulting in a decrease in the number of dynamic cross-linking sites, which loosens the originally compact cross-linking network and affects the tensile stress of the hydrogel. The maximum tensile lightness and fracture strain of the hydrogels increased with increasing Ag@MXene content. Among them, the maximum tensile stress in the hydrogel containing 3 wt% Ag@MXene was 2.056 KPa and 20.933% strain at break. The increase in mechanical properties is due to the fact that the surface of MXene in Ag@MXene nanosheets is rich in various functional groups, such as -OH and -O. The addition of a large amount of Ag@MXene forms hydrogen bonding interactions with PVA molecular chains, thus improving the mechanical properties of the hydrogel.

Then, different levels of HACC were added to determine the consistent strain at break of the 3 wt% Ag@M-PVA hydrogel with that of the pure PVA hydrogel. The effect of different levels of HACC on the mechanical properties of the Ag@M-H-PVA hydrogel was explored. In Figure 2b, the stress–strain plots of 0, 3 wt%, 6 wt%, and 9 wt% HACC composite hydrogels are displayed. The results clearly show that the addition of HACC increased the tensile strength of the hydrogels. Until the strain at break of 9 wt% HACC hydrogel reached 1266%, the tensile stress was 13.25 KPa. HACC participated in the cross-linking of several dynamic networks by strengthening the hydrogen bonding interaction between polymer PVA and Ag@MXene, promoting the diffusion of free radicals and the movement of molecular chains in curing, resulting in a dense Ag@M-H-PVA hydrogel cross-linked network, resulting in improved mechanical properties. Compared with pure PVA hydrogels, Ag@MXene-HACC-PVA hydrogels have multiple interactions between macromolecular chains and nanosheets, resulting in excellent mechanical properties.

### 3.3. Swelling Behavior

For biomedical applications, the swelling properties of hydrogels are crucial for their performance in biological environments. The swelling property is an important property of hydrogels that can affect aspects such as the water absorption capacity, mechanical properties, and biocompatibility of hydrogels. To investigate the swelling behavior of Ag@M-H-PVA hydrogels, we immersed the hydrogels in aqueous solutions and evaluated the swelling behavior of the hydrogels by measuring the mass before and after immersion. In Figure 3, the time–swelling rate of the composite hydrogel with different contents of HACC is compared in a graph. The composite hydrogels reached the swelling equilibrium at about 300 min. In the composite hydrogel with 3 wt% HACC added, the swelling rate was 6.69%. This is because the addition of HACC increases the density of the cross-linked network inside the hydrogel and increases the hydrogen bonding interaction, which makes the swelling capacity decrease. As the mass ratio of HACC increased, the hydrogel swelling rate also increased, and the swelling rate of the composite hydrogel containing 9 wt% HACC was 13%, but it was still lower than that of the Ag@MXene-PVA hydrogel without the addition of HACC. We believe that too much HACC loosens the hydrogel structure, leading to easier loosening of the macromolecular chains. This is consistent with the results of previous researchers [33].

### 3.4. Adhesion Properties

Adhesion is also an important property in medical hydrogel applications. The adhesion photos of the Ag@M-H-PVA hydrogel, as shown in the figure, demonstrate the hydrogel on skin, rubber, plastic, glass, paper, and other materials, and the adhesion strength is calculated by dividing the measured maximum load value by the area of the adhered sample, as shown in Figure 4a. It can be seen that the hydrogel can adhere to the surface of the human body and other materials and can bear a certain weight. Since there are abundant active functional groups in the HACC molecule, such as -OH, -NH, and -NH_2_, this allows the hydrogel itself to interact with other materials, thus achieving excellent adhesion properties.

Subsequently, the adhesion strength of the hydrogel surface was measured using a high- and low-temperature electronic tension machine. As a hard backing for the hydrogel, the hydrogel was adhered to the PET film using a cyanoacrylate adhesive. The peel test speed of the samples was set at 10 mm/s. As shown in Figure 4c, it was clearly found that the adhesion ability of the Ag@M-H-PVA hydrogels gradually increased with increasing content of HACC. Among them, the hydrogel with 9 wt% HACC had an adhesion as high as 8.8 ± 0.12 KPa; the hydrogel with 6 wt% HACC had an adhesion of 5.378 ± 0.1 KPa; the hydrogel with 3 wt% HACC had an adhesion of 3.022 ± 0.1 KPa; and the comparison group without HACC had an adhesion close to 0. The results indicated that HACC could effectively improve the hydrogel adhesion. This is beneficial for biomaterials, which can promote cell adhesion and transfer. As shown in Appendix A, the adhesion strength of the hydrogel was stabilized after 60 min under wet conditions, reaching 4.3 KPa, and the value was reduced by half. The hydrogel adhesion strength after 120 min under wet conditions reached 3.87 KPa. This indicates that the hydrogel can achieve some adhesion under wet conditions. The results indicate that it is beneficial for use in biomedical applications.

### 3.5. Electronic Behavior and Its Applications

The presence of nanosilver and MXene nanosheets imparts electrical conductivity to the hydrogel. As shown in Figure 5a, the conductivity of the hydrogel increased from 0 to 0.0082 ± 0.0001 S/cm with increasing Ag@MXene content, but, as shown in Figure 5b, different contents of HACC were added to the Ag@M-PVA hydrogel, and the conductivity of the hydrogel decreased from 0.0082 ± 0.0001 S/cm to 0.001215 ± 0.001 s/cm as the HACC content increased. This is due to the fact that the addition of HACC decreased the concentration of Ag@MXene in the whole system and hindered the movement of the silver nanoparticles and MXene nanosheets in the cross-linked network, resulting in a decrease in conductivity.

In addition to this, Ag@M-H-PVA hydrogels have high strain sensitivity and excellent stability. The hydrogels were connected to the LCR in order to obtain the real-time resistance during the strain of the hydrogels. Figure 5c–f show the relative resistance changes of the hydrogels (H6%, Ag@M3%) under certain conditions: wrist, finger, throat, and balloon bending, for example. The strain-sensing hydrogel showed good strain response to various motions. After two finger joint bending motions, the corresponding resistance changes remained almost constant. When the hydrogel was directly adhered to the throat, the mouth uttered “ah” three times to accurately detect the strain in the throat during speech. This further confirms the feasibility of Ag@M-H-PVA hydrogel as a strain sensor.

To better evaluate the sensing capability of the Ag@M-H-PVA hydrogel, the relative resistance change of the hydrogel at 0–110% tensile strain was recorded as in Figure 5g. The strain response curves reflect the tensile strain at three different slopes. The GF is 1.2721 at 0–40% tensile strain, 1.7617 at 40–90% tensile strain, and 1.251 at 90–110% tensile strain, showing good strain sensing capability.

### 3.6. In Vitro Antibacterial Activity

To confirm the antibacterial activity of the Ag@M-PVA hydrogel, the inhibition of the hydrogel after co-culture with Gram-positive Staphylococcus aureus and Gram-negative Escherichia coli was assessed using plate coating counts. The graphs of the inhibitory effect of the Ag@M-PVA and Ag@M-H-PVA hydrogels on different bacteria with/without 808 nm NIR irradiation are shown in Figure 6a–h. The Ag@M-PVA hydrogel was incubated with *E. coli* suspension and the strain reached 51.78% inhibition (Figure 6a), indicating that PVA was not significantly toxic and that the coincidence of Ag@MXene showed significant inhibition of *E. coli*, which is consistent with the results of previously published research [8]. Comparing Figure 6a,e shows that the growth inhibition of the Ag@M-PVA hydrogel on *E. coli* under 808 nm NIR irradiation increased significantly, and the inhibition rate increased from 51.78% to 97.5%. Comparing Figure 6b,f shows that the inhibition rate increased by 9.64% after irradiation. This also confirms that Ag@MXene has a synergistic photothermal antibacterial effect. AgNPs are reported to exert powerful bactericidal effects by interacting with bacterial membranes, causing structural damage, membrane depolarization, and downstream metabolic effects [34]. MXene nanosheets cut the bacterial cell wall and then enter the bacterial cytoplasm, releasing the bacterial DNA and eventually dispersing the bacteria. It was found that the smaller the size of MXene, the more damage it caused to the cell, especially under prolonged exposure [35].

To further investigate whether the addition of HACC improves the antibacterial activity of the hydrogels, observations and counts were performed using the same method. Comparing Figure 6a,c, the inhibition of *E. coli* increased from 51.78% to 87.14% after the addition of HACC without NIR light irradiation. This indicates that the addition of HACC can effectively improve the growth inhibition of *E. coli* by the hydrogel. Although the hydrogel was originally inhibited by 87.14% in the presence of HACC and Ag@MXene, some *E. coli* would still survive. After 808 nm NIR light irradiation, AgNPs and MXene nanosheets played a photothermal synergistic inhibition effect, which greatly reduced the number of *E. coli* colonies. After 15 min of NIR light irradiation (Figure 6g), the inhibition rate reached 99.56%. As shown in Figure 6d,h, Ag@M-H-PVA hydrogels with/without NIR light irradiation were almost free of colonies, resulting in the inability of colonies to continue to multiply. This indicates that HACC has a strong inhibitory effect on the proliferation of Staphylococcus aureus. This is because HACC increased the permeability of outer and inner membranes, allowing easy release of cell contents. This may be caused by the electrostatic interaction between the NH_3_^+^ group of HACC and the phosphorylation group of the phospholipid fraction of the cell membrane [36]. The results showed that the addition of HACC effectively improved the antibacterial activity of the hydrogel.

### 3.7. In Vitro Biocompatibility

The safety of hydrogels as biomaterials is essential to study. The relative cell viability changes of L929 with the PVA hydrogel, Ag@MXene-PVA hydrogel, and Ag@M-H-PVA hydrogel infusions at 1 d and 3 d were examined using the CCK-8 method. As shown in Figure 7a, the relative viability changes of L929 cells after co-incubation of the PVA hydrogel, Ag@M-PVA hydrogel, and Ag@M-H-PVA hydrogel extracts with L929 cells for 1 d and 3 d were above 85%. Preliminarily, we confirmed that the content of each component was within a reasonable range and did not produce cytotoxicity. We also assessed the biocompatibility by the live/dead staining method. Figure 7b shows images of laser confocal microscopy observation after co-incubation of the extracts with cells, and the images show live cells in green. There was no significant difference in the number of cells after 1 d of co-incubation compared to the control group, and the cells showed a perfect spindle shape under the microscope. Over time, after 3 d of co-incubation, the images showed that the components had no significant effect on the survival of L929 cells. This is consistent with the results of the CCK-8 method described above for detecting changes in relative cell viability. The results indicate that the hydrogels are safe and non-toxic and can be applied in the field of biomaterials.

## 4. Conclusions

In conclusion, this study reports that 2-hydroxypropyltrimethylammonium chloride chitosan and Ag@MXene can be used to prepare double-network hydrogels by acetal reaction using PVA as a substrate. Successful polymerization was confirmed by SEM, EDS, and FTIR spectrograms. As a result, a strain-sensing antibacterial composite hydrogel is proposed: under 808 nm NIR light irradiation, the composite hydrogel showed significant inhibition of Gram-negative *E. coli* and *S. aureus* by photothermal synergy. Cellular experiments confirmed the good biocompatibility of Ag@MXene-HACC-PVA hydrogels. Thus, this study shows that Ag@MXene-HACC-PVA hydrogels have potential as biomedical materials.

## Figures and Tables

**Figure 1 polymers-15-02352-f001:**
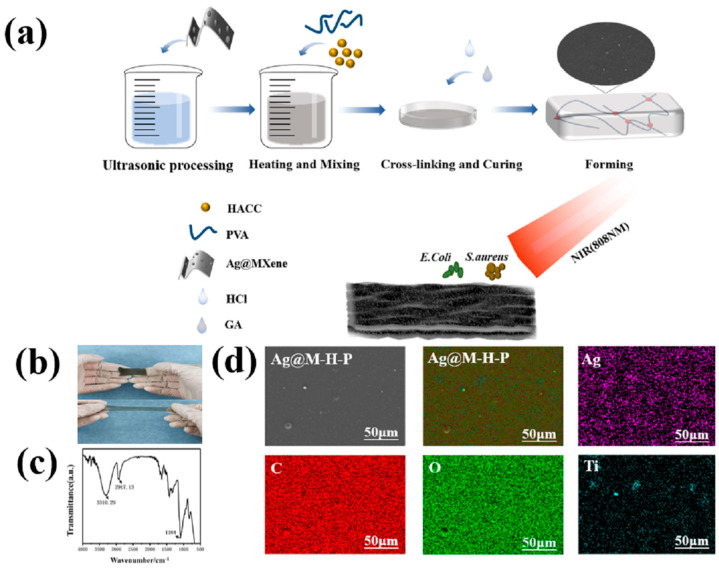
(**a**) Ag@MXene−H−PVA hydrogel preparation process; (**b**) Ag@MXene−H−PVA hydrogel FT−IR spectrum; (**c**) hydrogel stretching before and after comparison chart; (**d**) SEM image of hydrogel and its surface EDS analysis.

**Figure 2 polymers-15-02352-f002:**
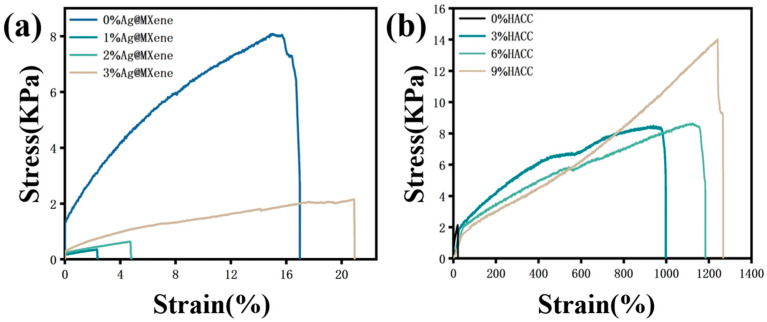
(**a**) Stress–strain curves of Ag@MXene-PVA hydrogels with different Ag@MXene contents. (**b**) Stress–strain curves of Ag-H-PVA hydrogels with different contents of HACC.

**Figure 3 polymers-15-02352-f003:**
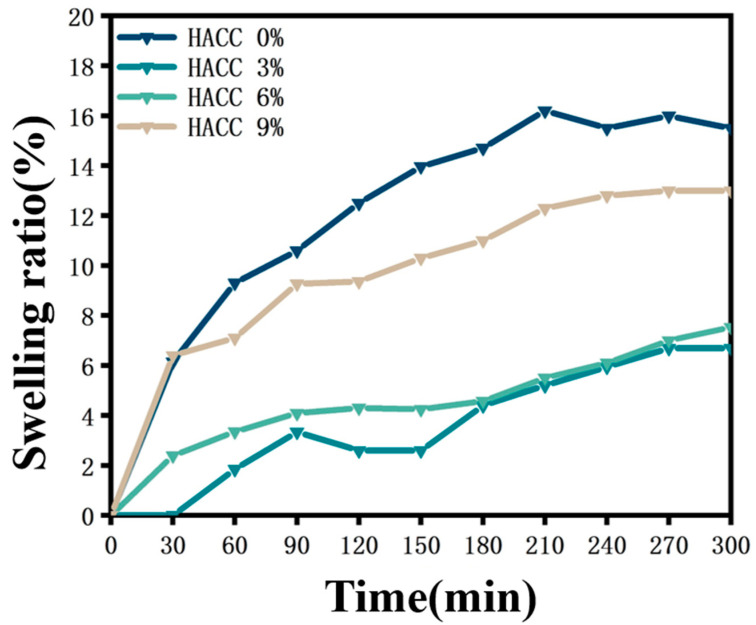
Variation curve of swelling rate–time of Ag@M-H-PVA composite hydrogel.

**Figure 4 polymers-15-02352-f004:**
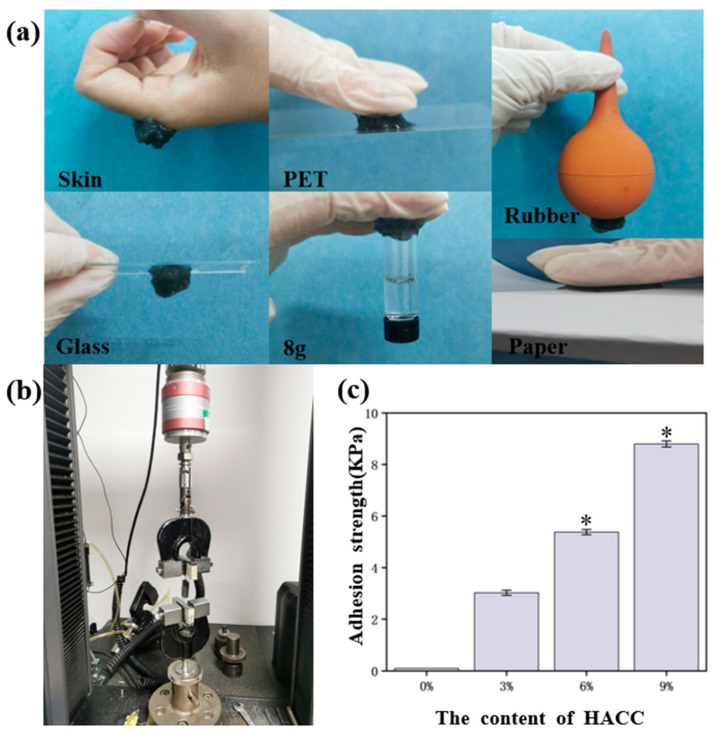
(**a**) Demonstration of Ag@M-H-PVA hydrogel adhesion to skin, rubber, plastic, glass, paper, etc.; (**b**) demonstration chart when measuring adhesion strength; (**c**) adhesion strength of different contents of HACC; * *p* < 0.05.

**Figure 5 polymers-15-02352-f005:**
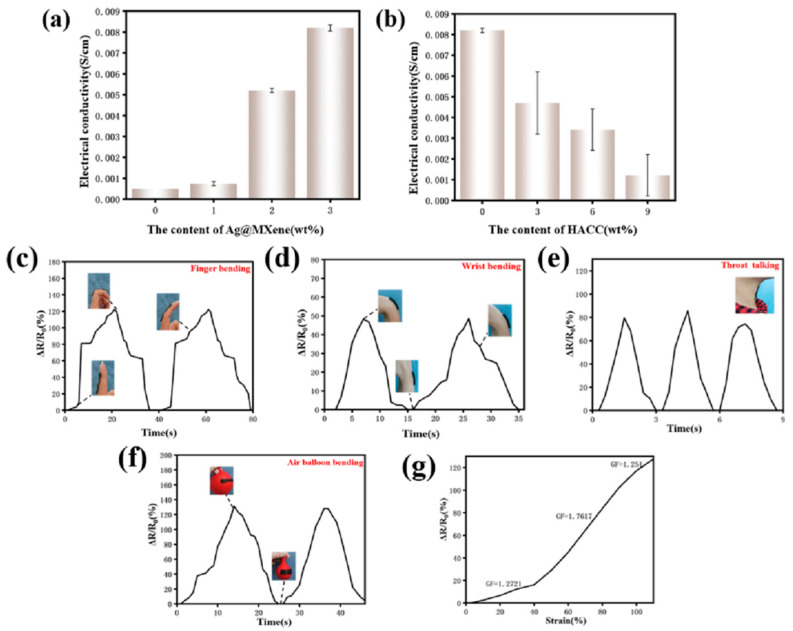
(**a**) Conductivity diagram of Ag@MXene-PVA hydrogel with different levels of Ag@MXene; (**b**) conductivity diagram of Ag@M-H-P hydrogel with different levels of HACC; (**c**–**f**) relative resistance changes of Ag@M-H-PVA hydrogel under various motion tests: finger, wrist, throat, and balloon; (**g**) relative resistance changes of Ag@M-H-PVA hydrogels at different tensile strains (0~110%).

**Figure 6 polymers-15-02352-f006:**
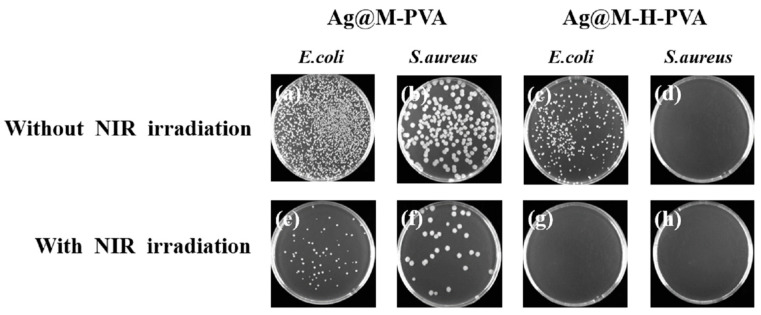
Photograph of *E. coli* and *S. aureus* colonies on a plate. (**a**,**c**) Ag@M-PVA and Ag@M-H-PVA groups that did not receive 808 nm NIR irradiation with *E. coli*; (**b**,**d**) Ag@M-PVA and Ag@M-H-PVA groups that did not receive 808 nm NIR irradiation with S. aureus; (**e**,**g**) Ag@M-PVA and Ag@M-H-PVA groups that received 808 nm NIR irradiation with *E. coli*; (**f**,**h**) Ag@M-PVA and Ag@M-H-PVA groups that received 808 nm NIR irradiation with *S. aureus*.

**Figure 7 polymers-15-02352-f007:**
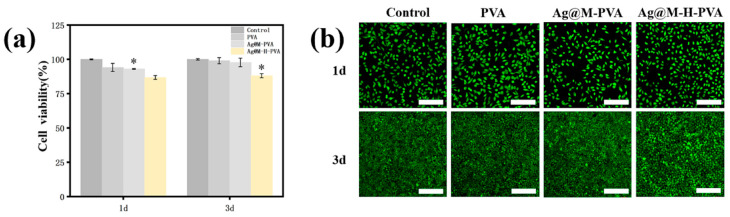
Biocompatibility evaluation of hydrogels. (**a**) Relative cell viability of PVA, Ag@MXene-PVA, and Ag@MXene-HACC-PVA. (**b**) Fluorescence images of PVA, Ag@MXene-PVA, and Ag@MXene-HACC-PVA after 1 day and 3 days of incubation for L929 live/dead assay; * *p* < 0.05. Scale bar is 100 μm.

## Data Availability

Data sharing is not applicable to this article.

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
