# Peer review of "Antimicrobial and Mechanical Properties of Ag@Ti3C2Tx-Modified PVA Composite Hydrogels Enhanced with Quaternary Ammonium Chitosan"

_polymers, 2023, doi:10.3390/polym15102352_

Round 1

Reviewer 1 Report

This work reports a hybrid hydrogel composed of PVA, Ag@MXene, and 2-hydroxypro-pyltrimethylammonium chloride chitosan (HACC). With adding HACC, the gel becomes strong, ductile, and has excellent adhesion. In addition, the gel has some functions, including conductive, antibacterial, and biocompatibility. In general, this work is a systematic study of new materials. It is recommended to publish on Polymers. However, before publication, the authors are recommended to address the following comments:

  1. In Fig. 2a, why do the 1% and 3% Ag@MXene-PVA gels significantly differ in fracture strain? And the 3% Ag@MXene-PVA gel shows almost similar fracture strain with the pure PVA. As the authors say,” the addition of Ag@MXene breaks the dynamic equilibrium within the original hydrogel, resulting in a decrease in the number of dynamic crosslinking sites”, from this point of view, it is difficult to follow why the fracture strain of 1% Ag@MXene-PVA gel is much smaller than the 3% Ag@MXene-PVA gel.
  2. In Fig. 2b, the authors compared the tensile behaviors of the gels with varied HACC contents. How about the tensile behavior of pure HACC gel? Whether 9% HACC can become a gel?
  3. In Fig. 3, the authors say “This is because the addition of HACC increases the density of the cross-linked network inside the hydrogel and increases the hydrogen bonding interaction, which makes the swelling capacity decrease.” More HACC should form more hydrogen bonding to act as crosslinker, and decrease the swelling ratio. So why does the HACC 9% show the largest swelling ratio compared with the other gels containing HACC?
  4. In Fig. 7b, it seems that the cells change their shape after 3 days of incubation compared with the 1 day’s incubation. What is the scar bar?
  5. Please recheck each sentence. Some sentences have no punctuation, and some are repeated, for example, on Page 2, lines 59 and 71.

Author Response

We thank the reviewer for their valuable time and comments. Please see the attachment.

Reviewer 2 Report

This manuscript presented a study on the mechanical and antimicrobial properties of a Ag@MXene-H-PVA composite hydrogel. The material demonstrated good cell viability, good antimicrobial properties and adhesion. The authors also demonstrated the sensing capability of hydrogel.

1.     PVA can be made into exceptionally tough materials. In this manuscript, the mechanical performance is on the lower side. The authors used a high content of GA to crosslink PVA, have the authors tried different GA contents? Have the authors considered thermal post treatments to improve mechanical properties?

2.     The authors showed good adhesion properties of the material. To serve for applications in living tissue, adhesion under wet conditions will be favored. Does the material presented maintain adhesion when the material surface is wetted?

Minor editing of English language required

Author Response

We thank the reviewer for his/her valuable time and comments. Please see the attachment.
